# Just-In-Time Learning for Fast and Flexible Inference

**S. M. Ali Eslami, Daniel Tarlow, Pushmeet Kohli and John Winn**
Microsoft Research
{alie,dtarlow,pkohli,jwinn}@microsoft.com

## Abstract

Much of research in machine learning has centered around the search for inference algorithms that are both general-purpose and efficient. The problem is extremely challenging and general inference remains computationally expensive. We seek to address this problem by observing that in most specific applications of a model, we typically only need to perform a small subset of all possible inference computations. Motivated by this, we introduce *just-in-time learning*, a framework for fast and flexible inference that learns to speed up inference at run-time. Through a series of experiments, we show how this framework can allow us to combine the flexibility of sampling with the efficiency of deterministic message-passing.

## 1 Introduction

We would like to live in a world where we can define a probabilistic model, press a button, and get accurate inference results within a matter of seconds or minutes. Probabilistic programming languages allow for the rapid definition of rich probabilistic models to this end, but they also raise a crucial question: what algorithms can we use to efficiently perform inference for the largest possible set of programs in the language? Much of recent research in machine learning has centered around the search for inference algorithms that are both flexible and efficient.

The general inference problem is extremely challenging and remains computationally expensive. Sampling based approaches (*e.g.* [5, 19]) can require many evaluations of the probabilistic program to obtain accurate inference results. Message-passing based approaches (*e.g.* [12]) are typically faster, but require the program to be expressed in terms of functions for which efficient message-passing operators have been implemented. However, implementing a message-passing operator for a new function either requires technical expertise, or is computationally expensive, or both.

In this paper we propose a solution to this problem that is automatic (it doesn't require the user to build message passing operators) and efficient (it learns from past experience to make future computations faster). The approach is motivated by the observation that general algorithms are solving problems that are harder than they need to be: in most *specific* inference problems, we only ever need to perform a small subset of all possible message-passing computations. For example, in Expectation Propagation (EP) the range of input messages to a logistic factor, for which it needs to compute output messages, is highly problem specific (see Fig. 1a). This observation raises the central question of our work: can we automatically speed up the computations required for general message-passing, at run-time, by learning about the statistics of the specific problems at hand?

Our proposed framework, which we call just-in-time learning (JIT learning), initially uses highly general algorithms for inference. It does so by computing messages in a message-passing algorithm using Monte Carlo sampling, freeing us from having to implement hand-crafted message update operators. However, it also gradually learns to increase the speed of these computations by regressing from input to output messages (in a similar way to [7]) at run-time. JIT learning enables us to combine the *flexibility* of sampling (by allowing arbitrary factors) and the *speed* of hand-crafted message-passing operators (by using regressors), without having to do any pre-training. This constitutes our main contribution and we describe the details of our approach in Sec. 3.

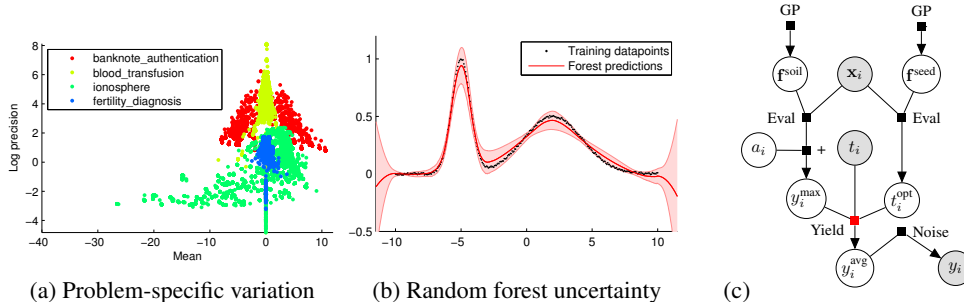

(a) Problem-specific variation  (b) Random forest uncertainty  (c)

Figure 1: (a) Parameters of Gaussian messages input to a logistic factor in logistic regression vary significantly in four random UCI datasets. (b) Figure for Sec. 4: A regression forest performs 1D regression (1,000 trees, 2 feature samples per node, maximum depth 4, regressor polynomial degree 2). The red shaded area indicates one standard deviation of the predictions made by the different trees in the forest, indicating its uncertainty. (c) Figure for Sec. 6: The yield factor relates temperatures and yields recorded at farms to the optimal temperatures of their planted grain. JIT learning enables us to incorporate arbitrary factors with ease, whilst maintaining inference speed.

Our implementation relies heavily on the use of regressors that are aware of their own uncertainty. Their awareness about the limits of their knowledge allows them to decide when to trust their predictions and when to fall back to computationally intensive Monte Carlo sampling (similar to [8] and [9]). We show that random regression forests [4] form a natural and efficient basis for this class of 'uncertainty aware' regressors and we describe how they can be modified for this purpose in Sec. 4. To the best of our knowledge this is the first application of regression forests to the self-aware learning setting and it constitutes our second contribution.

To demonstrate the efficacy of the JIT framework, we employ it for inference in a variety of graphical models. Experimental results in Sec. 6 show that for general graphical models, our approach leads to significant improvements in inference speed (often several orders of magnitude) over importance sampling whilst maintaining overall accuracy, even boosting performance for models where hand designed EP message-passing operators are available. Although we demonstrate JIT learning in the context of expectation propagation, the underlying ideas are general and the framework can be used for arbitrary inference problems.

## 2 Background

A wide class of probabilistic models can be represented using the framework of *factor graphs*. In this context a factor graph represents the factorization of the joint distribution over a set of random variables $\mathbf{x} = \{x_1, ..., x_V\}$ via non-negative factors $\psi_1, ..., \psi_F$ given by $p(\mathbf{x}) = \prod_f \psi_f(\mathbf{x}_{\mathrm{ne}(\psi_f)})/Z$, where $\mathbf{x}_{\mathrm{ne}(\psi_f)}$ is the set of variables that factor $\psi_f$ is defined over. We will focus on directed factors of the form $\psi(\mathbf{x}_{\mathrm{out}}|\mathbf{x}_{\mathrm{in}})$ which directly specify the conditional density over the output variables $\mathbf{x}_{\mathrm{out}}$ as a function of the inputs $\mathbf{x}_{\mathrm{in}}$, although our approach can be extended to factors of arbitrary form.

Belief propagation (or sum-product) is a message-passing algorithm for performing inference in factor graphs with discrete and real-valued variables, and it includes sub-routines that compute variable-to-factor and factor-to-variable messages. The bottleneck is mainly in computing the latter kind, as they often involve intractable integrals. The message from factor $\psi$ to variable $i$ is:

$$m_{\psi \to i}(x_i) = \int_{\mathbf{x}_{-i}} \psi(\mathbf{x}_{\mathrm{out}}|\mathbf{x}_{\mathrm{in}}) \prod_{k \in \mathrm{ne}(\psi) \setminus i} m_{k \to \psi}(x_k), \tag{1}$$

where $\mathbf{x}_{-i}$ denotes all random variables in $\mathbf{x}_{\mathrm{ne}(\psi)}$ except $i$. To further complicate matters, the messages are often not even representable in a compact form. Expectation Propagation [11] extends the applicability of message-passing algorithms by projecting messages back to a pre-determined, tractable family distribution:

$$m_{\psi \to i}(x_i) = \frac{\mathrm{proj}\left[\int_{\mathbf{x}_{-i}} \psi(\mathbf{x}_{\mathrm{out}}|\mathbf{x}_{\mathrm{in}}) \prod_{k \in \mathrm{ne}(\psi)} m_{k \to \psi}(x_k)\right]}{m_{i \to \psi}(x_i)}. \tag{2}$$

The proj[·] operator ensures that the message is a distribution of the correct type and only has an effect if its argument is outside the approximating family used for the target message.

The integral in the numerator of Eq. 2 can be computed using Monte Carlo methods [2, 7], *e.g.* by using the generally applicable technique of importance sampling. After multiplying and dividing by a proposal distribution $q(\mathbf{x}_{\text{in}})$ we get:

$$m_{\psi \to i}(x_i) \equiv \text{proj}\left[ \int_{\mathbf{x}_{-i}} v(\mathbf{x}_{\text{in}}, \mathbf{x}_{\text{out}}) \cdot w(\mathbf{x}_{\text{in}}, \mathbf{x}_{\text{out}}) \right] / m_{i \to \psi}(x_i), \tag{3}$$

where $v(\mathbf{x}_{\text{in}}, \mathbf{x}_{\text{out}}) = q(\mathbf{x}_{\text{in}})\psi(\mathbf{x}_{\text{out}}|\mathbf{x}_{\text{in}})$ and $w(\mathbf{x}_{\text{in}}, \mathbf{x}_{\text{out}}) = \prod_{k \in \text{ne}(\psi)} m_{k \to \psi}(x_k)/q(\mathbf{x}_{\text{in}})$. Therefore

$$m_{\psi \to i}(x_i) \simeq \text{proj}\left[ \frac{\sum_s w(\mathbf{x}_{\text{in}}^s, \mathbf{x}_{\text{out}}^s)\delta(x_i)}{\sum_s w(\mathbf{x}_{\text{in}}^s, \mathbf{x}_{\text{out}}^s)} \right] / m_{i \to \psi}(x_i), \tag{4}$$

where $\mathbf{x}_{\text{in}}^s$ and $\mathbf{x}_{\text{out}}^s$ are samples from $v(\mathbf{x}_{\text{in}}, \mathbf{x}_{\text{out}})$. To sample from $v$, we first draw values $\mathbf{x}_{\text{in}}^s$ from $q$ then pass them through the forward-sampling procedure defined by $\psi$ to get a value for $\mathbf{x}_{\text{out}}^s$.

Crucially, note that we require no knowledge of $\psi$ other than the ability to sample from $\psi(\mathbf{x}_{\text{out}}|\mathbf{x}_{\text{in}})$. This allows the model designer to incorporate arbitrary factors simply by providing an implementation of this forward sampler, which could be anything from a single line of deterministic code to a large stochastic image renderer. However, drawing a single sample from $\psi$ can itself be a time-consuming operation, and the complexity of $\psi$ and the arity of $\mathbf{x}_{\text{in}}$ can both have a dramatic effect on the number of samples required to compute messages accurately.

## 3   Just-in-time learning of message mappings

Monte Carlo methods (as defined above) are computationally expensive and can lead to slow inference. In this paper, we adopt an approach in which we *learn* a direct mapping, parameterized by $\boldsymbol{\theta}$, from variable-to-factor messages $\{m_{k \to \psi}\}_{k \in \text{ne}(\psi)}$ to a factor-to-variable message $m_{\psi \to i}$:

$$\overline{m}_{\psi \to i}(x_i) \equiv f(\{m_{k \to \psi}\}_{k \in \text{ne}(\psi)}|\boldsymbol{\theta}). \tag{5}$$

Using this direct mapping function $f$, factor-to-variable messages can be computed in a fraction of the time required to perform full Monte Carlo estimation. Heess *et al.* [7] recently used neural networks to learn this mapping offline for a broad range of input message combinations.

Motivated by the observation that the distribution of input messages that a factor sees is often problem specific (Fig. 1a), we consider learning the direct mapping *just-in-time* in the context of a specific model. For this we employ 'uncertainty aware' regressors. Along with each prediction $\overline{m}$, the regressor produces a scalar measure $\overline{u}$ of its uncertainty about that prediction:

$$\overline{u}_{\psi \to i} \equiv u(\{m_{k \to \psi}\}_{k \in \text{ne}(\psi)}|\boldsymbol{\theta}). \tag{6}$$

We adopt a framework similar to that of *uncertainty sampling* [8] (also [9]) and use these uncertainties at run-time to choose between the regressor's estimate and slower 'oracle' computations:

$$m_{\psi \to i}(x_i) = \begin{cases} \overline{m}_{\psi \to i}(x_i) & \overline{u}_{\psi \to i} < u^{\text{max}} \\ m_{\psi \to i}^{\text{oracle}}(x_i) & \text{otherwise} \end{cases} \tag{7}$$

where $u^{\text{max}}$ is the maximum tolerated uncertainty for a prediction. In this paper we consider importance sampling or hand-implemented Infer.NET operators as oracles however other methods such as MCMC-based samplers could be used. The regressor is updated after every oracle consultation in order to incorporate the newly acquired information.

An appropriate value for $u^{\text{max}}$ can be found by collecting a small number of Monte Carlo messages for the target model offline: the uncertainty aware regressor is trained on some portion of the collected messages, and evaluated on the held out portion, producing predictions $\overline{m}_{\psi \to i}$ and confidences $\overline{u}_{\psi \to i}$ for every held out message. We then set $u^{\text{max}}$ such that no held out prediction has an error above a user-specified, problem-specific maximum tolerated value $D^{\text{max}}$.

A natural choice for this error measure is mean squared error of the parameters of the messages (*e.g.* natural parameters for the exponential family), however this is sensitive to the particular parameterization chosen for the target distribution type. Instead, for each pair of predicted and oracle messages

from factor $\psi$ to variable $i$, we calculate the marginals $\bar{b}_i$ and $b_i^{\text{oracle}}$ they each induce on the target random variable, and compute the Kullback-Leibler (KL) divergence between the two:

$$D_{\text{KL}}^{\text{mar}}(\overline{m}_{\psi \to i} \| m_{\psi \to i}^{\text{oracle}}) \equiv D_{\text{KL}}(\bar{b}_i \| b_i^{\text{oracle}}), \qquad (8)$$

where $\bar{b}_i = \overline{m}_{\psi \to i} \cdot m_{i \to \psi}$ and $b_i^{\text{oracle}} = m_{\psi \to i}^{\text{oracle}} \cdot m_{i \to \psi}$, using the fact that beliefs can be computed as the product of incoming and outgoing messages on any edge. We refer to the error measure $D_{\text{KL}}^{\text{mar}}$ as *marginal KL* and use it throughout the JIT framework, as it encourages the system to focus efforts on the quantity that is ultimately of interest: the accuracy of the posterior marginals.

## 4    Random decision forests for JIT learning

We wish to learn a mapping from a set of incoming messages $\{m_{k \to \psi}\}_{k \in \text{ne}(\psi)}$ to the outgoing message $m_{\psi \to i}$. Note that separate regressors are trained for each outgoing message. We require that the regressor: 1) trains and predicts efficiently, 2) can model arbitrarily complex mappings, 3) can adapt dynamically, and 4) produces uncertainty estimates. Here we describe how decision forests can be modified to satisfy these requirements. For a review of decision forests see [4].

In EP, each incoming and outgoing message can be represented using only a few numbers, *e.g.* a Gaussian message can be represented by its natural parameters. We refer to the outgoing message by $m_{\text{out}}$ and to the set of incoming messages by $\mathbf{m}_{\text{in}}$. Each set of incoming messages $\mathbf{m}_{\text{in}}$ is represented in two ways: the first, a concatenation of the parameters of its constituent messages which we call the 'regression parameterization' and denote by $\mathbf{r}_{\text{in}}$; and the second, a vector of features computed on the set which we call the 'tree parameterization' and denote by $\mathbf{t}_{\text{in}}$. This tree parametrization typically contains values for a larger number of properties of each constituent message (*e.g.* parameters *and* moments), and also properties of the set as a whole (*e.g.* $\psi$ evaluated at the mode of $\mathbf{m}_{\text{in}}$). We represent the outgoing message $m_{\text{out}}$ by a vector of real valued numbers $\mathbf{r}_{\text{out}}$. Note that $d_{\text{in}}$ and $d_{\text{out}}$, the number of elements in $\mathbf{r}_{\text{in}}$ and $\mathbf{r}_{\text{out}}$ respectively, need not be equal.

**Weak learner model.** Data arriving at a split node $j$ is separated into the node's two children according to a binary weak learner $h(\mathbf{t}_{\text{in}}, \boldsymbol{\tau}_j) \in \{0, 1\}$, where $\boldsymbol{\tau}_j$ parameterizes the split criterion. We use weak learners of the generic oriented hyperplane type throughout (see [4] for details).

**Prediction model.** Each leaf node is associated with a subset of the labelled training data. During testing, a previously unseen set of incoming messages traverses the tree until it reaches a leaf which by construction is likely to contain similar training examples. We therefore use the statistics of the data gathered in that leaf to predict outgoing messages with a multivariate polynomial regression model of the form: $\mathbf{r}_{\text{out}}^{\text{train}} = \mathbf{W} \cdot \phi^n(\mathbf{r}_{\text{in}}^{\text{train}}) + \epsilon$, where $\phi^n(\cdot)$ is the $n$-th degree polynomial basis function, and $\epsilon$ is the $d_{\text{out}}$-dimensional vector of normal error terms. We use the learned $d_{\text{out}} \times d_{\text{in}}$-dimensional matrix of coefficients $\mathbf{W}$ at test time to make predictions $\bar{\mathbf{r}}_{\text{out}}$ for each $\mathbf{r}_{\text{in}}$. To recap, $\mathbf{t}_{\text{in}}$ is used to traverse message sets down to leaves, and $\mathbf{r}_{\text{in}}$ is used by the linear regressor to predict $\mathbf{r}_{\text{out}}$.

**Training objective function.** The optimization of the split functions proceeds in a greedy manner. At each node $j$, depending on the subset of the incoming training set $\mathcal{S}_j$ we learn the function that 'best' splits $\mathcal{S}_j$ into the training sets corresponding to each child, $\mathcal{S}_j^{\text{L}}$ and $\mathcal{S}_j^{\text{R}}$, *i.e.* $\boldsymbol{\tau}_j = \text{argmax}_{\boldsymbol{\tau} \in \mathcal{T}_j} I(\mathcal{S}_j, \boldsymbol{\tau})$. This optimization is performed as a search over a discrete set $\mathcal{T}_j$ of a random sample of possible parameter settings. The number of elements in $\mathcal{T}_j$ is typically kept small, introducing random variation in the different trees in the forest. The objective function $I$ is:

$$I(\mathcal{S}_j, \boldsymbol{\tau}) = -E(\mathcal{S}_j^{\text{L}}, \mathbf{W}^{\text{L}}) - E(\mathcal{S}_j^{\text{R}}, \mathbf{W}^{\text{R}}), \qquad (9)$$

where $\mathbf{W}^{\text{L}}$ and $\mathbf{W}^{\text{R}}$ are the parameters of the polynomial regression models corresponding to the left and right training sets $\mathcal{S}_j^{\text{L}}$ and $\mathcal{S}_j^{\text{R}}$, and the 'fit residual' $E$ is:

$$E(\mathcal{S}, \mathbf{W}) = \frac{1}{2} \sum_{\mathbf{m}_{\text{in}} \in \mathcal{S}} D_{\text{KL}}^{\text{mar}}(\overline{m}_{\mathbf{m}_{\text{in}}}^{\mathbf{W}} \| m_{\mathbf{m}_{\text{in}}}^{\text{oracle}}) + D_{\text{KL}}^{\text{mar}}(m_{\mathbf{m}_{\text{in}}}^{\text{oracle}} \| \overline{m}_{\mathbf{m}_{\text{in}}}^{\mathbf{W}}). \qquad (10)$$

Here $\mathbf{m}_{\text{in}}$ is a set of incoming messages in $\mathcal{S}$, $m_{\mathbf{m}_{\text{in}}}^{\text{oracle}}$ is the oracle outgoing message, $\overline{m}_{\mathbf{m}_{\text{in}}}^{\mathbf{W}}$ is the estimate produced by the regression model specified by $\mathbf{W}$ and $D_{\text{KL}}^{\text{mar}}$ is the marginal KL. In simple terms, this objective function splits the training data at each node in a way that the relationship between the incoming and outgoing messages is well captured by the polynomial regression in each child, as measured by symmetrized marginal KL.

**Ensemble model.** A key aspect of forests is that their trees are randomly different from each other. This is due to the relatively small number of weak learner candidates considered in the optimization of the weak learners. During testing, each test point $\mathbf{m}_{\text{in}}$ simultaneously traverses all trees from their roots until it reaches their leaves. Combining the predictions into a single forest prediction may be done by averaging the parameters $\overline{\mathbf{r}}_{\text{out}}^t$ of the predicted outgoing messages $\overline{m}_{\text{out}}^t$ by each tree $t$, however again this would be sensitive to the parameterizations of the output distribution types. Instead, we compute the *moment average* $\overline{m}_{\text{out}}$ of the distributions $\{\overline{m}_{\text{out}}^t\}$ by averaging the first few moments of each predicted distribution across trees, and solving for the distribution parameters which match the averaged moments. Grosse *et al.* [6] study the characteristics of the moment average in detail, and have showed that it can be interpreted as minimizing an objective function $\overline{m}_{\text{out}} = \text{argmin}_m U(\{\overline{m}_{\text{out}}^t\}, m)$ where $U(\{\overline{m}_{\text{out}}^t\}, m) = \sum_t D_{\text{KL}}(\overline{m}_{\text{out}}^t \| m)$.

Intuitively, the level of agreement between the predictions of the different trees can be used as a proxy of the forest's uncertainty about that prediction (we choose not to use uncertainty within leaves in order to maintain high prediction speed). If all the trees in the forest predict the same output distribution, it means that their knowledge about the function $f$ is similar *despite* the randomness in their structures. We therefore set $\overline{u}_{\text{out}} \equiv U(\{\overline{m}_{\text{out}}^t\}, \overline{m}_{\text{out}})$. A similar notion is used for classification forests, where the entropy of the aggregate output histogram is used as a proxy of the classification's uncertainty [4]. We illustrate how this idea extends to simple regression forests in Fig. 1b, and in Sec. 6 we also show empirically that this uncertainty measure works well in practice.

**Online training.** During learning, the trees periodically obtain new information in the form of $(\mathbf{m}_{\text{in}}, m_{\text{out}}^{\text{oracle}})$ pairs. The forest makes use of this by pushing $\mathbf{m}_{\text{in}}$ down a portion $0 < \rho \leq 1$ of the trees to their leaf nodes and retraining the regressors at those leaves. Typically $\rho = 1$, however we use values smaller than 1 when the trees are shallow (due to the mapping function being captured well by the regressors at the leaves) and the forest's randomness is too low to produce reliable uncertainty estimates. If the regressor's fit residual $E$ at a leaf (Eq. 10) is above a user-specified threshold value $E_{\text{leaf}}^{\text{max}}$, a split is triggered on that node. Note that no depth limit is ever specified.

## 5 Related work

There are a number of works in the literature that consider using regressors to speed up general purpose inference algorithms. For example, the *Inverse MCMC* algorithm [20] uses discriminative estimates of local conditional distributions to make proposals for a Metropolis-Hastings sampler, however these predictors are not aware of their own uncertainty. Therefore the decision of when the sampler can start to rely on them needs to be made manually and the user has to explicitly separate offline training and test-time inference computations.

A related line of work is that of inference machines [14, 15, 17, 13]. Here, message-passing is performed by a sequence of predictions, where the sequence itself is defined by the graphical model. The predictors are jointly trained to ensure that the system produces correct labellings, however the resulting inference procedure no longer corresponds to the original (or perhaps to any) graphical model and therefore the method is unsuitable if we care about querying the model's latent variables.

The closest work to ours is [7], in which Heess *et al.* use neural networks to learn to pass EP messages. However, their method requires the user to anticipate the set of messages that will ever be sent by the factor ahead of time (itself a highly non-trivial task), and it has no notion of confidence in its predictions and therefore it will silently fail when it sees unfamiliar input messages. In contrast the JIT learner trains in the context of a specific model thereby allocating resources more efficiently, and because it knows what it knows, it buys generality without having to do extensive pre-training.

## 6 Experiments

We first analyze the behaviour of JIT learning with diagnostic experiments on two factors: logistic and compound gamma, which were also considered by [7]. We then demonstrate its application to a challenging model of US corn yield data. The experiments were performed using the extensible factor API in Infer.NET [12]. Unless stated otherwise, we use default Infer.NET settings (*e.g.* for message schedules and other factor implementations). We set the number of trees in each forest to 64 and use quadratic regressors. Message parameterizations and graphical models, experiments on a product factor and a quantitative comparison with [7] can be found in the supplementary material.

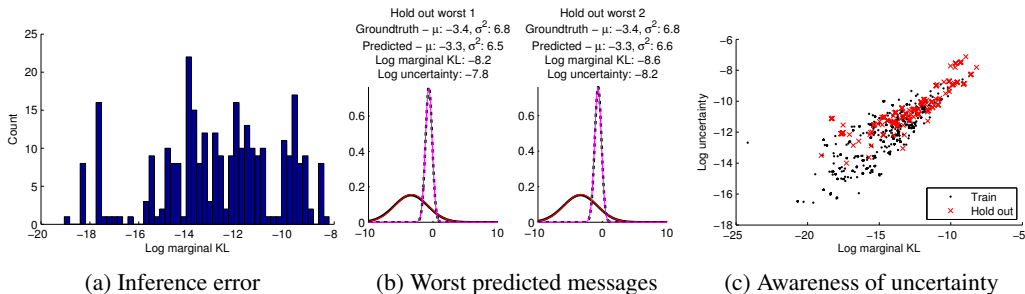

(a) Inference error      (b) Worst predicted messages      (c) Awareness of uncertainty

Figure 2: **Uncertainty aware regression.** All plots for the Gaussian forest. (a) Histogram of marginal KLs of outgoing messages, which are typically very small. (b) The forest's most inaccurate predictions (black: $m^{\text{oracle}}$, red: $\overline{m}$, dashed black: $b^{\text{oracle}}$, purple: $\overline{b}$). (c) The regressor's uncertainty increases in tandem with marginal KL, *i.e.* it does not make confident but inaccurate predictions.

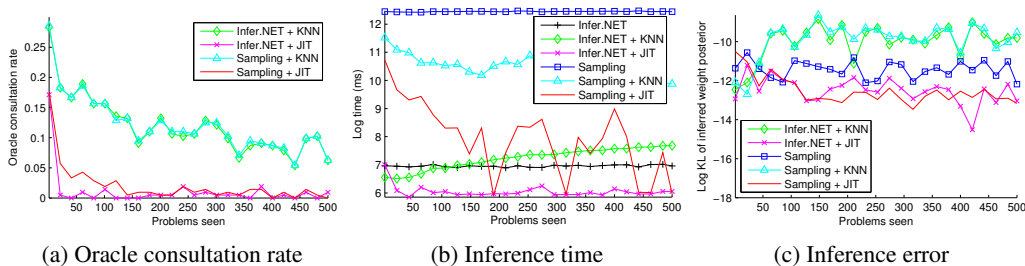

(a) Oracle consultation rate      (b) Inference time      (c) Inference error

Figure 3: **Logistic JIT learning.** (a) The factor consults the oracle for only a fraction of messages, (b) leading to significant savings in time, (c) whilst maintaining (or even decreasing) inference error.

**Logistic.** We have access to a hand-crafted EP implementation of this factor, allowing us to perform quantitative analysis of the JIT framework's performance. The logistic deterministically computes $x_{\text{out}} = \sigma(x_{\text{in}}) = 1/(1+\exp\{-x_{\text{in}}\})$. Sensible choices for the incoming and outgoing message types are Gaussian and Beta respectively. We study the logistic factor in the context of Bayesian logistic regression models, where the relationship between an input vector $\mathbf{x}$ and a binary output observation $y$ is modeled as $p(y = 1) = \sigma(\mathbf{w}^T\mathbf{x})$. We place zero-mean, unit-variance Gaussian priors on the entries of regression parameters $\mathbf{w}$, and run EP inference for 10 iterations.

We first demonstrate that the forests described in Sec. 4 are fast and accurate uncertainty aware regressors by applying them to five synthetic logistic regression 'problems' as follows: for each problem, we sample a groundtruth $\mathbf{w}$ and training $\mathbf{x}$s from $\mathcal{N}(0, 1)$ and then sample their corresponding $y$s. We use a Bayesian logistic regression model to infer $\mathbf{w}$s using the training datasets and make predictions on the test datasets, whilst recording the messages that the factor receives and sends during both kinds of inference. We split the observed message sets into training (70%) and hold out (30%), and train and evaluate the random forests using the two datasets. In Fig. 2 we show that the regressor is accurate and that it is uncertain whenever it makes predictions with higher error.

One useful diagnostic for choosing the various parameters of the forests (including choice of parametrization for $\mathbf{r}_{\text{in}}$ and $\mathbf{t}_{\text{in}}$, as well leaf tolerance $E_{\text{leaf}}^{\text{max}}$) is the average utilization of its leaves during held out prediction, *i.e.* what fraction of leaves are visited at test time. In this experiment the forests obtain an average utilization of 1, meaning that every leaf contributes to the predictions of the 30% held out data, thereby indicating that the forests have learned a highly compact representation of the underlying function. As described in Sec. 3, we also use the data gathered in this experiment to find an appropriate value of $u^{\text{max}}$ for use in just-in-time learning.

Next we evaluate the uncertainty aware regressor in the context of JIT learning. We present several related regression problems to a JIT logistic factor, *i.e.* we keep $\mathbf{w}$ fixed and generate multiple new $\{(\mathbf{x}, y)\}$ sets. This is a natural setting since often in practice we observe multiple datasets which we believe to have been generated by the same underlying process. For each problem, using the JIT factor we infer the regression weights and make predictions on test inputs, comparing wall-clock time and accuracy with non-JIT implementations of the factor. We consider two kinds of oracles:

those that consult Infer.NET's message operators and those that use importance sampling (Eq. 4). As a baseline, we also implemented a $K$-nearest neighbour (KNN) uncertainty aware regressor. Here, messages are represented using their natural parameters, the uncertainty associated with each prediction is the mean distance from the $K$-closest points in this space, and the outgoing message's parameters are found by taking the average of the parameters of the $K$-closest output messages. We use the same procedure as the one described in Sec. 3 to choose $u^{\max}$ for KNN.

We observe that the JIT factor does indeed learn about the inference problem over time. Fig. 3a shows that the rate at which the factor consults the oracle decreases over the course of the experiment, reaching zero at times (*i.e.* for these problems the factor relies entirely on its predictions). On average, the factor sends 97.7% of its messages without consulting the sampling oracle (a higher rate of 99.2% when using Infer.NET as the oracle, due to lack of sampling noise), which leads to several orders of magnitude savings in inference time (from around 8 minutes for sampling to around 800 ms for sampling + JIT), even increasing the speed of our Infer.NET implementation (from around 1300 ms to around 800 ms on average, Fig. 3b). Note that the forests are not merely memorising a mapping from input to output messages, as evidenced by the difference in the consultation rates of JIT and KNN, and that KNN speed deteriorates as the database grows. Surprisingly, we observe that the JIT regressors in fact decrease the KL between the results produced by importance sampling and Infer.NET, thereby increasing overall inference accuracy (Fig. 3c, this could be due to the fact that the regressors at the leaves of the forests smooth out the noise of the sampled messages). Reducing the number of importance samples to reach speed parity with JIT drastically degrades the accuracy of the outgoing messages, increasing overall log KL error from around $-11$ to around $-4$.

**Compound gamma.** The second factor we investigate is the compound gamma factor. The compound gamma construction is used as a heavy-tailed prior over precisions of Gaussian random variables, where first $r_2$ is drawn from a gamma with rate $r_1$ and shape $s_1$ and the precision of the Gaussian is set to be a draw from a gamma with rate $r_2$ and shape $s_2$. Here, we have access to closed-form implementations of the two gamma factors in the construction, however we use the JIT framework to collapse the two into a single factor for increased speed.

We study the compound gamma factor in the context of Gaussian fitting, where we sample a random number of points from multiple Gaussians with a wide range of precisions, and then infer the precision of the generating Gaussians via Bayesian inference using a compound gamma prior. The number of samples varies between 10 and 100 and the precision varies between $10^{-4}$ and $10^4$ in each problem. The compound factor learns the message mapping after around 20 problems (see Fig. 4a). Note that only a single message is sent by the factor in each episode, hence the abrupt drop in inference time. This increase in performance comes at negligible loss of accuracy (Figs. 4b, 4c).

**Yield.** We also consider a more realistic application to scientific modelling. This is an example of a scenario for which our framework is particularly suited: scientists often need to build large models with factors that directly take knowledge about certain components of the problem into account. We use JIT learning to implement a factor that relates agriculture yields to temperature in the context of an ecological climate model. Ecologists have strong empirical beliefs about the form of the relationship between temperature and yield (that yield increases gradually up to some optimal temperature but drops sharply after that point; see Fig 5a and [16, 10]) and it is imperative that this relationship is modelled faithfully. Deriving closed form message-operators is a non-trivial task, and therefore current state-of-the-art is sampling-based (*e.g.* [3]) and highly computationally intensive.

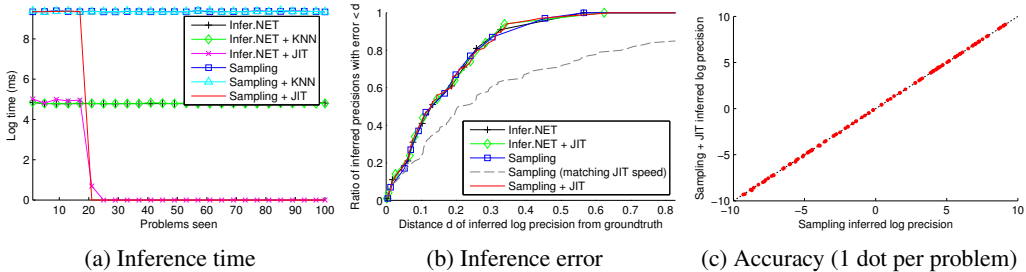

(a) Inference time      (b) Inference error      (c) Accuracy (1 dot per problem)

Figure 4: **Compound gamma JIT learning.** (a) JIT reduces inference time for sampling from $\sim$11 seconds to $\sim$1 ms. (b) JIT s posteriors agree highly with Infer.NET. Using fewer samples to match JIT speed leads to degradation of accuracy. (c) Increased speed comes at negligible loss of accuracy.

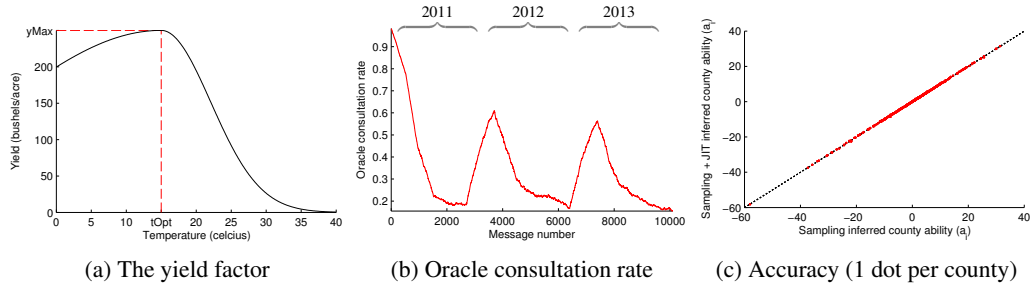

<div align="center">
(a) The yield factor      (b) Oracle consultation rate      (c) Accuracy (1 dot per county)
</div>

Figure 5: **A probabilistic model of corn yield.** (a) Ecologists believe that yield increases gradually up to some optimal temperature but drops sharply after that point [16, 10], and they wish to incorporate this knowledge into their models faithfully. (b) Average consultation rate per 1,000 messages over the course of inference on the three datasets. Notice decrease within and across datasets. (c) Significant savings in inference time (Table 1) come at a small cost in inference accuracy.

We obtain yield data for 10% of US counties for 2011–2013 from the USDA National Agricultural Statistics Service [1] and corresponding temperature data using [18]. We first demonstrate that it is possible to perform inference in a large-scale ecological model of this kind with EP (graphical model shown in Fig. 1c; derived in collaboration with computational ecologists; see supplementary material for a description), using importance sampling to compute messages for the yield factor for which we lack message-passing operators. In addition to the difficulty of computing messages for the multidimensional yield factor, inference in the model is challenging as it includes multiple Gaussian processes, separate $t^{\mathrm{opt}}$ and $y^{\mathrm{max}}$ variables for each location, many copies of the yield factor, and its graph is loopy. Results of inference are shown in the supplementary material.

We find that with around 100,000 samples the message for the yield factor can be computed accurately, making these by far the slowest computations in the inference procedure. We apply JIT learning by regressing these messages instead. The high arity of the factor makes the task particularly challenging as it increases the complexity of the mapping function being learned. Despite this, we find that when performing inference on the 2011 data the factor can learn to accurately send up to 54% of messages without having to consult the oracle, resulting in a speedup of 195%.

A common scenario is one in which we collect more data and wish to repeat inference. We use the forests learned at the end of inference on 2011 data to perform inference on 2012 data, and the forests learned at the end of this to do inference on 2013 data, and compare to JIT learning from scratch for each dataset. The factor transfers its knowledge across the problems, increasing inference speedup from 195% to 289% and 317% in the latter two experiments respectively (Table 1), whilst maintaining overall inference accuracy (Fig. 5c).

|  | IS | JIT fresh | | JIT continued | |
|---|---|---|---|---|---|
|  | Time | FR | Speedup | FR | Speedup |
| 11 | 451s | 54% | 195% | — | — |
| 12 | 449s | 54% | 192% | 60% | 288% |
| 13 | 451s | 54% | 191% | 64% | 318% |

Table 1: FR is fraction of regressions with no oracle consultation.

## 7 Discussion

The success of JIT learning depends heavily on the accuracy of the regressor and its knowledge about its uncertainty. Random forests have shown to be adequate however alternatives may exist, and a more sophisticated estimate of uncertainty (*e.g.* using Gaussian processes) is likely to lead to an increased rate of learning. A second critical ingredient is an appropriate choice of $u^{\mathrm{max}}$, which currently requires a certain amount of manual tuning.

In this paper we showed that it is possible to speed up inference by combining EP, importance sampling and JIT learning, however it will be of interest to study other inference settings where JIT ideas might be applicable. Surprisingly, our experiments also showed that JIT learning can increase the accuracy of sampling or accelerate hand-coded message operators, suggesting that it will be fruitful to use JIT to remove bottlenecks even in existing, optimized inference code.

**Acknowledgments**

Thanks to Tom Minka and Alex Spengler for valuable discussions, and to Silvia Caldararu and Drew Purves for introducing us to the corn yield datasets and models.

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
