[Supplementary Material]

# Just-In-Time Learning
# for Fast and Flexible Inference
## Supplementary Material

# 1 Parameterizations of messages

## 1.1 Tree parameterization

The tree parameterization of a message set $\mathbf{t}_{\text{in}}$ consists of the concatenation of:

1. All common parameterizations of each distribution type. For a Gaussian this includes mean, variance, mean-times-precision and precision, for a beta it includes $\alpha$, $\beta$, mean and variance and for a gamma it includes rate, shape, scale, mean and variance,

2. Binary features indicating whether the messages are proper, uniform or point-masses, and

3. The value of the $\psi$ evaluated at the mode of $\boldsymbol{\mu}_{\text{in}}$.

The tree parameterization can be computed automatically for each factor and is a function of the types of the distributions of the incoming messages.

## 1.2 Regression parameterization

The regression parameterization of messages for each factor is chosen for each problem with numerical stability and convenience in mind. Gaussian messages are parameterized using their means and log-precisions, so that messages with high or even infinite variance are represented accurately. For beta messages, log-$\alpha$ and log-$\beta$ parameterizations work best in practice, and for gamma messages log-shape and log-rate parameterizations work best.

Figure 1: **Logistic regression.** Factor graph for the logistic regression model. The logistic factor is highlighted in red. JIT learning is used to compute messages outgoing from the factor. $\mathbf{x}_i$ and $\mathbf{w}_i \in \mathbb{R}^D$, $z_i = \mathbf{w}_i^T \mathbf{x}_i$, $p_i = \sigma(z_i)$, and $y_i \sim \text{Bernoulli}(p_i)$, *i.e.* $y_i \in \{0, 1\}$.

Figure 2: **Comparison with Heess *et al.***  JIT achieves good accuracy using few training examples and a low test-time consultation rate. Heess et al. is also able to achieve good accuracy, but it requires approximately an order of magnitude more oracle consultations in order to do so.

## 2   Comparison with Heess *et al.*

The closest work to ours is [1], in which Heess *et al.* use neural networks to learn to pass EP messages. They train the neural networks independently of any given model and therefore attack a problem that is harder than it needs to be.

Heess *et al.* can be arbitrarily accurate or inaccurate depending on the quality and volume of the synthetic data provided by the user for pre-training. When comparing the two methods, one must take into account a multi-way trade-off between computation invested into pre-training (only for Heess et al.), oracle computation invested at test time (only for JIT), and the amount of error that can be tolerated for the application at hand.

We have produced one such comparison for the logistic factor, where we plot the performance of just-in-time learning vs. Heess *et al.* style pre-training (but using forests) with varying amounts of data (see Fig. 2). There are 200 red points corresponding to JIT performance after seeing each problem. The blue curve corresponds to the average performance of the different Heess factors on the 200 problems. For first, middle and last problems, we report JITs consultation rate, which in turn gives an indication of the computation invested at test time.

JIT achieves good accuracy using few training examples and a low test-time consultation rate. Heess et al.  is also able to achieve good accuracy, but it requires approximately an order of magnitude more oracle consultations in order to do so.

By training in the context of a specific model, JIT learning allocates resources more efficiently and also side-steps the problem of artificially creating training data for the regressors, which can be particularly expensive for high-arity factors. Because the JIT learner knows what it knows, it buys generality without having to do extensive pre-training.

## 3   Gaussian product factor

We also applied JIT learning to the Gaussian product factor, which computes $x_{\text{out}} = x_{\text{in}}^1 \times x_{\text{in}}^2$. Here, all choices for the incoming and outgoing message types are Gaussian. The Gaussian product is a highly challenging factor to work with in EP. Reasons include symmetries in the behaviour of the factor due to signs of input messages, and the fact that the message outputs can change very quickly as functions of message inputs. Heess *et al.* report that they were unable to learn a general, stable neural network implementation of this factor [1].

(a) Prediction uncertainty       (b) Prediction accuracy       (c) Tree depth

Figure 3: **Just-in-time learning of the Gaussian product factor.** (a) Trace of forest uncertainty $\bar{u}_{\mathrm{out}}$ during inference for the final 10 problems in a single run (boundaries of the 10 problems visualised by vertical bars at the top of the figure). Red dots indicate predictions whose uncertainty was above $u^{\mathrm{max}}$, leading to an oracle consultation. (b) The learned factor's predictions agree highly with the built-in factor. (c) Sampling noise leads to deeper trees.

Figure 4: **Multiplicative noise regression.** Factor graph for the multiplicative noise regression model. JIT learning is used to compute messages outgoing from the factor highlighted in red. $\mathbf{x}_i$ and $\mathbf{w}_i \in \mathbb{R}^D$, $z_i = \mathbf{w}_i^T \mathbf{x}_i$, $p_i = 1 + \epsilon_i$, and $y_i = z_i \times p_i$, i.e. $y_i \in \mathbb{R}$.

Again, we observe that in many specific models that the Gaussian product appears in, it only ever needs to perform a small subset of all message-passing computations. For example in a multiplicative noise regression of the form $y = (\mathbf{w}^T \mathbf{x}) \times (1 + \epsilon)$, where $\epsilon$ is Gaussian with a small variance, the second input to the product factor always takes on values around 1. We experiment with JIT learning of a product factor in this class of models (see Fig. 4). As before we present a JIT Gaussian product factor several regression problems, keeping $\mathbf{w}$ fixed and generating new $\{(\mathbf{x}, y)\}$ sets. For each problem, we infer the regression weights and make predictions on test inputs.

We observe that JIT learning is capable of learning a performant Gaussian product factor for this problem from importance sampling (Fig. 3), reducing inference time using sampling from around 22 seconds to around 3 seconds whilst maintaining good accuracy. However, at a given precision, the consultation rate drops at a slower speed than for the logistic or compound gamma, indicating the difficulty of learning a regressor for this function that generalises accurately. This is also evidenced by the larger average depth of trees in the forest (Fig. 3c) and the lower recorded leaf utilization rate of 0.84 (at 30% hold-out). By choosing a different parametrization for the messages or a different family of predictors at the leaves it may be possible to increase the forest's performance.

(a) Yield posteriors      (b) Optimal temperature mean      (c) Optimal temperature variance

Figure 5: **A probabilistic model of corn yield.** Ecologists believe that yield increases gradually up to some optimal temperature but drops sharply after that point [3, 2], and they wish to incorporate this knowledge into their models faithfully. (a) Posterior means and 1 standard deviations of county abilities $a_i$, maximum corn yields $y_i^{\mathrm{max}}$ and observed values of corn yield $y_i$ for a random selection of locations. (b,c) Posterior mean and variance on the optimal temperature $t^{\mathrm{opt}}$. This information is used by ecologists to identify counties that are using sub-optimal grain varieties. Black dots indicate locations of counties for which we collect data.

# 4   The crop yield model

We use a graphical model to capture the relationship between farm $i$'s 2D position $x_i$, its yield $y_i$ and its temperature $t_i$. From empirical studies, we know that yield increases gradually up to some optimal temperature $t^{\mathrm{opt}}$ but drops sharply after that point (see [3, 2]):

$$y_i^{\mathrm{avg}} = \begin{cases} y_i^{\mathrm{max}} \cdot \exp\{-p^{\mathrm{low}} \cdot (t_i - t^{\mathrm{opt}})^2\} & t_i < t^{\mathrm{opt}} \\ y_i^{\mathrm{max}} \cdot \exp\{-p^{\mathrm{high}} \cdot (t_i - t^{\mathrm{opt}})^2\} & \text{otherwise,} \end{cases} \tag{1}$$

where $p^{\mathrm{low}} = 10^{-3}$ and $p^{\mathrm{high}} = 10^{-2}$. We incorporate this knowledge into the model using a JIT factor. We assume that $t^{\mathrm{opt}}$ varies smoothly over space (e.g. due to nearby farms having access to similar, but slightly different, seed varieties) and encode it into the model using a Gaussian Process. At their optimal temperatures, different farms operate at different yields (e.g. due to varying qualities of soil). This quantity also varies smoothly over space but at a different lengthscale. We additionally introduce county-level biases $a_i$ to account for varying levels of farming experience in different counties.

We observe county positions $\mathbf{x}_i$ (longitude and latitude) and temperatures $t_i$, and infer their maximum yields $y_i^{\mathrm{max}}$ and the optimal operating temperature of their grain $t_i^{\mathrm{opt}}$. This information is used by ecologists *e.g.* to identify counties that are using sub-optimal grain varieties.