[Reviews · NeurIPS 2014]

Submitted by Assigned_Reviewer_12

This paper is an extension of the recent Heess et al "Learning to Pass Expectation Propagation Messages". AIUI, Heess et al uses neural networks to learn an approximate mapping between input and output messages. In Hees et al, learning is done generically for each model, before specific problem instances are seen. Their approach has the drawback of requiring the compilation of synthetic training data, and being unable to adapt to model peculiarities during the message-passing phase. The present paper fixes both of these problems by learning input-output mappings as messages are being passed. It does this by falling back to an "oracle" for each message until the approximate mapping at that node has been suitably trained. The oracle is simply the (presumably expensive) way in which messages would have been computed in the absence of an approximation.

The approach is tested on two synthetic problems, as well as on a probabilistic model of corn yield. Infer.NET is used throughout.

The paper presents some important concepts, and I think it should be accepted because it is a "good enough" presentation of concepts which will have to be published at some point anyway. My main criticism is that the experiments seem biased in favor of the proposed method. For instance, the last subsection of Experiments says "We find that with around 100,000 samples the message for the yield factor can be computed accurately, making these by far the slowest computations in the inference procedure". The factor in Figure 5a looks pretty simple, I don't see why some sort of smooth approximation couldn't be used to compute messages going through it - 100,000 samples sounds excessive. At the same time, the proposed method for learning approximate message-operators based on random decision forests seems very slow. Perhaps another reviewer who is more familiar with expectation propagation and Infer.NET could comment on this.

Perhaps also because I don't have a good background on expectation propagation, I find that after reading the paper I don't have a satisfactory intuition about why the proposed method, as well as the one in Heess et al, is needed. Do messages really jump around so much as they converge, that we need neural networks or random decision forests to speed this up? It would be nice if there was room to go into more detail about what's happening under the hood. Or am I missing something.

Minor comments: "Regressor" should be defined, in Wikipedia it is defined as "independent variable", but I think the meaning is different in this paper. Expectation propagation (and Infer.NET?) should be mentioned in the abstract! "Related work" is usually Section 2, I am not sure why this paper puts it at Section 5.

Clarity: The paper is fairly clear. Important details of experiments and models have been moved to Supplementary Material, which makes it a bit harder to understand what is happening in detail.

Originality: The paper is original. It is to a certain degree an "engineering" paper, in that a number of different approximate techniques (EP, importance sampling, random forests) are drawn together without full justification. But I think this comes with the territory.

I think I addressed quality and significance above.
Summary: I was interested by the paper, and glad to be able to read and review it. However, after reading I still had some questions about the applicability of the method and the fairness of the experiments.

Submitted by Assigned_Reviewer_25

The paper proposes an online framework to increase the computational efficiency of generic message passing algorithms which utilize e.g. sampling to determine the output messages from non-analytical factors. The key novelty is that random decision forests are used in an online (Just-In¬-Time) fashion to learn problem-specific mappings from factor input messages to output messages. Subsequent message-passing iterations can utilize these learned mappings within and across datasets to avoid unnecessary and potentially time-consuming (tilted) factor moment integrations. By experimental comparisons using both simulated and real-world data, the authors demonstrate that the framework can achieve significant savings in the number of factor integrations compared with regular sample-based or analytical message passign algorithms.

Quality: The paper is technically sound and well structured, and the methodology is developed in a solid manner relying on various existing techniques and references to related research. The approach relies on some heuristic choices but I found the authors’ justification for these sensible, and the approach seems to work well in practice. The following points could be clarified to further improve the presentation:

1) I didn’t understand why the JIT learning is able to increase upon the accuracy of regular sampling-based message passing although the approximate families are the same. The authors noted this observation but I couldn’t find their explanation for this?

2) I couldn't find the exact mathematical description of the crop-yield factor from the supplementary material? Based on the factor graph, it seems to me that the outgoing messages could be computed using either 2- or 3- dimensional integrations? If so, how would the JIT learning work with factors that require very high dimensional integrations?

3) It would be interesting to see how the JIT learning compares performance-wise with respect to the offline approach of Heess et al.

Clarity: The paper reads well and I enjoyed reading it. I found only the section 4 a bit hard to follow: If space permits, it would be nice to describe very shortly the basic idea and the overall usage for random decision forests around line 188 before going in to the details. Also in lines 199-215 I could quickly deduce if the coefficients W are re-estimated at each node split decisions.

Originality: The idea of learning the factor input-output mappings has already been proposed in offline and non-dataset-specific manner using neural networks as pointed out by the authors. However, using computationally cheaper random forest in online fashion and utilizing the dataset specific structure of the messages is a very nice idea that seems to work well in practice, and is worthy of publication in my opinion.

Significance: According to the experiments, the proposed method can significantly decrease the computational burden of general sampling based message passing algorithms in both simulated and real-world modeling problems. Also the generality of the approach could offer useful insights for other inference settings. Therefore I would like to see the paper published.
Summary: The paper proposes an online framework that uses simple but efficient techniques to learn problem-specific input-output message mappings to decrease the computational burden of generic sampling-based message passing algorithms. According to the experiments, the proposed approach can give significant savings in the number of potentially time-consuming factor integrations.

Submitted by Assigned_Reviewer_36

Belief propagation (BP) is a widely-used algorithm for performing inference. Dealing with continuous variables is a challenge, since computing the messages involves integration which is a time-consuming operation. In this paper the authors suggest learning a function that maps the variables’ messages to a factor message and thereby “learn” the integration. Having this mapping, calculation of the integral occurs only rarely.

The setting of the paper or the exact problem they are solving is not well-defined.
The advantage of this paper over the paper by Heess et al needs to be made more clear.
The authors use the term “inference” for calculating factor messages - I think it is confusing.
The authors use the term “training” (for example lines 235, 310). The main point of the article is that no training is necessary, so the use of training here is unclear.
In the experiments section, comparison to other methods is not presented. Comparison to Heess et al and other methods is necessary.
Can the authors say somthing about the use of their method in the convergence of BP? Since they cluster messages, how does it affect the convergence of BP?
In the experiments, figure 3.b - why is there initially a small drop when using only infer.Net? Moreover, when using infer.Net+JIT, the rate initially is shown as slower than when using infer.Net alone (and should initially be faster, and only afterwards slow and intersect)- why?
Was the u^{max} pre-trained? If so, this is an illegal advantage for your algorithm (did the authors do the same for KNN?).
What is meant by the legend “problems seen”. The same model but different evidence? Same model structure with different parameters? Only the same factor type?
Can one learn messages in a single run between different iterations of BP?
Summary: I think that having the work of Heess a justification for this work will be clear only with through evaluation and comparison between different methods.
Author Feedback
Author rebuttal: We thank the reviewers for their time and useful comments. We will incorporate all of the writing suggestions. Here we clarify the key points raised in the reviews:

R3: What is the exact problem the method is solving?

Implementing a message-passing operator (for any new factor type) either requires technical expertise, or is computationally expensive, or both. This is undesirable in the probabilistic programming setting, where technical expertise and computational resources are limited. The proposed solution is automatic (it requires little knowledge of message-passing from the user) and it is fast (it gradually learns about the task required of it until passes messages very quickly).

R3: What is the advantage of JIT over Heess et al.?

Heess et al. tackle the same problem. But, their method 1) requires the user to anticipate the set of messages that will ever be sent by the factor ahead of time (itself a highly non-trivial task), and 2) has no notion of confidence in its predictions and will silently fail when it sees unfamiliar input messages. JIT learning does not require specifying the factor’s domain in advance, and it provides a confidence associated with each prediction, allowing it to consult the oracle to maintain accuracy.

R3 and R2: How does JIT’s performance compare to Heess et al.?

Heess et al. can be arbitrarily accurate or inaccurate depending on the quality and volume of the synthetic data provided by the user for pre-training. When comparing the two methods, one must take into account a multi-way trade-off between computation invested into pre-training (only for Heess et al.), oracle computation invested at test time (only for JIT learning), and the amount of error that can be tolerated for the application at hand. Note that our aim has /not/ been to compare forests and neural nets, which to some extent are implementation details in this setting.

At the reviewer’s suggestion, we have produced one such comparison for the logistic factor (http://bit.ly/1ouI12g), where we plot the performance of just-in-time learning vs. Heess et al. style pre-training (but using forests) with varying amounts of data. The y axis represents inference error in the regression setting described in Sec. 6 (log KL of around -9 can be tolerated in this application).

There are 200 red points corresponding to JIT performance after seeing each problem. The blue curve corresponds to the average performance of the different Heess factors on the 200 problems. For first, middle and last problems, we report JIT’s consultation rate, which in turn gives an indication of the computation invested at test time.

JIT achieves good accuracy using few training examples and a low test-time consultation rate. Heess et al. is also able to achieve good accuracy, but it requires approximately an order of magnitude more oracle consultations in order to do so. We will add this plot to the final paper.

R1: Do messages really jump around so much as they converge that we need random forests to speed up computations? Are 100,000 samples really needed to compute messages accurately?

Unfortunately, the answer is 'yes' to both questions. Message-passing can move through unexpected intermediate stages before arriving at the final answer, and we cannot afford to send inaccurate messages at any stage, as this could lead to divergence or have a large negative effect on overall inference accuracy. BP can be fairly brittle in this regard.

Also note that the samples are not used to approximate the function depicted in Fig. 5a, but to compute the multi-dimensional integral in the numerator of Eq. 2. When the messages m_{k -> \psi} ‘disagree’ strongly (i.e. they put probability mass on x_k in a way that doesn’t ‘make sense’ according to the potential function \psi), a large number of samples may be required to compute the integral accurately using importance sampling. Note that outgoing messages from a factor can also change very quickly as functions of the incoming messages (e.g. see Fig. 3 of Heess et al.).

We illustrate some of these points in Fig. 4b (grey). Using fewer samples to match JIT speed leads to significant degradation of accuracy.

Other questions:

R2: Why is JIT more accurate than sampling-based message-passing?

We believe this to be due to the fact that the regressors at the leaves of the forests smooth out the noise of the sampled message estimates to a limited extent.

R2: Are the W coefficients re-estimated at each node split decision?

Yes.

R3: Why is there a small dip when using Infer.NET alone?

The small bump turns out to be due an Infer.NET routine which compiles the inference code when it is first run. We will correct for this in the final paper. Note that this does not affect any of the conclusions drawn from the plots.

R3: Why is Infer.NET + JIT initially slower than Infer.NET alone?

Perhaps we are misunderstanding the reviewer’s comments, but in Fig 3b. Infer.NET + JIT (pink) is faster than Infer.NET alone (black) from the beginning.

R3: Is u^max selected for KNN in the same manner as for JIT?

Yes. We use exactly the same procedure as the one described in Sec. 3 (lines 155 to 158) to choose u^max for KNN. We will make this point clear in the final paper.

R3: Since JIT clusters messages, what is its effect on the convergence of BP?

It has no effect. The JIT factor aims to send exactly those messages that would normally be sent by the factor, only faster. We are unsure what the reviewer means by ‘clustering of messages’. However, we agree that it would be of interest to explore the possibility of using JIT to send messages that actively aim to accelerate convergence.

R3: Can messages be learned between different iterations of BP?

Yes, that is exactly what happens in all experiments. The just-in-time factor can (and in fact does) learn from every single iteration of inference (line 153).